# Decisional Conflict after Deciding on Potential Participation in Early Phase Clinical Cancer Trials: Dependent on Global Health Status, Satisfaction with Communication, and Timing

**DOI:** 10.3390/cancers14061500

**Published:** 2022-03-15

**Authors:** Liza G. G. van Lent, Maja J. A. de Jonge, Mirte van der Ham, Marjolein van Mil, Eelke H. Gort, Jeroen Hasselaar, Esther Oomen-de Hoop, Carin C. D. van der Rijt, Julia C. M. van Weert, Martijn P. Lolkema

**Affiliations:** 1Department of Medical Oncology, Erasmus MC Cancer Institute, Erasmus University Medical Center, 3015 GD Rotterdam, The Netherlands; m.dejonge@erasmusmc.nl (M.J.A.d.J.); m.vanderham@erasmusmc.nl (M.v.d.H.); e.oomen-dehoop@erasmusmc.nl (E.O.-d.H.); c.vanderrijt@erasmusmc.nl (C.C.D.v.d.R.); m.lolkema@erasmusmc.nl (M.P.L.); 2Department of Medical Oncology and Clinical Pharmacology, Antoni van Leeuwenhoek, The Netherlands Cancer Institute, 1066 CX Amsterdam, The Netherlands; m.v.mil@nki.nl; 3Department of Medical Oncology, UMC Utrecht Cancer Center, 3584 CX Utrecht, The Netherlands; e.h.gort-2@umcutrecht.nl; 4Department of Anesthesiology, Pain & Palliative Medicine, Radboud University Medical Center, 6525 GA Nijmegen, The Netherlands; jeroen.hasselaar@radboudumc.nl; 5Department of Communication Science, Amsterdam School of Communication Research (ASCoR), University of Amsterdam, 1018 WV Amsterdam, The Netherlands; j.c.m.vanweert@uva.nl

**Keywords:** early phase clinical trials, decision making, quality of life, patient satisfaction, health literacy, hope

## Abstract

**Simple Summary:**

Early phase clinical trials are an essential part of modern drug development and thus the advance of anti-cancer therapies for patients. However, deciding whether to participate in such trials can be complex and patients have reported decisional conflict (i.e., unresolved decisional needs). The aim of our study was to untangle several factors that contribute to decisional conflict in patients with advanced cancer who have recently been asked to decide whether to participate in early phase clinical trials. We found that patients experienced less decisional conflict if they had a better global health status, higher satisfaction, and made their decision sooner. Other factors, such as the decision to (not) participate, did not prove to be the best indicators for decisional conflict. With these insights, we can start to build hypotheses on how to improve the decision-making process for patients with end-stage cancer, which can ultimately improve their quality of life.

**Abstract:**

When standard treatment options are not available anymore, patients with advanced cancer may participate in early phase clinical trials. Improving this complex decision-making process may improve their quality of life. Therefore, this prospective multicenter study with questionnaires untangles several contributing factors to decisional conflict (which reflects the quality of decision-making) in patients with advanced cancer who recently decided upon early phase clinical trial participation (phase I or I/II). We hypothesized that health-related quality of life, health literacy, sense of hope, satisfaction with the consultation, timing of the decision, and the decision explain decisional conflict. Mean decisional conflict in 116 patients was 30.0 (*SD* = 16.9). Multivariate regression analysis showed that less decisional conflict was reported by patients with better global health status (β = −0.185, *p* = 0.018), higher satisfaction (β = −0.246, *p* = 0.002), and who made the decision before (β = −0.543, *p* < 0.001) or within a week after the consultation (β = −0.427, *p* < 0.001). These variables explained 37% of the variance in decisional conflict. Healthcare professionals should realize that patients with lower global health status and who need more time to decide may require additional support. Although altering such patient intrinsic characteristics is difficult, oncologists can impact the satisfaction with the consultation. Future research should verify whether effective patient-centered communication could prevent decisional conflict.

## 1. Introduction

Many patients with cancer ultimately exhaust their standard treatment options. If they are in sufficiently good condition, they can opt to participate in early phase clinical trials (i.e., experimental treatments for which no clinical evidence of benefit is yet available); alternatively, they can choose to withhold systemic therapy and be supported by palliative care. Since it is unclear whether early phase clinical trial participation results in a survival benefit, a well-balanced decision-making process is essential to ensure that these patients spend the limited time towards the end of their lives in line with their personal values and beliefs [1]. Moreover, improving the patient decision-making process is known to be beneficial for a health-related quality of life, including in patients with end-stage cancer [2,3]. Therefore, investigating the quality of the decision-making process on whether to participate in early phase clinical trials is important for optimizing quality of life for these end-stage cancer patients.

One of the most, and increasingly used, outcomes to evaluate the quality of decision-making is decisional conflict, usually measured with the well-validated Decisional Conflict Scale (DCS) [4,5]. It indicates the extent to which patients report unresolved decisional needs such as personal uncertainty and related deficits in knowledge, values clarity, and support or pressure [6]. The higher the score (0–100), the more the decisional needs are yet unresolved. Data from previous studies support that patients with scores above 25 display clinically relevant decisional conflict [5,7], meaning that these patients need help resolving remaining issues in their decision-making process. The DCS has been used regularly for patients with cancer [8], but predominantly for standard treatment decisions. An exhaustive scoping review indicated that DCS scores across cancer treatment decisions are slightly clinically elevated (around 28, derived from figure), whereas end of life/palliative care decisions show (one of) the highest levels of decisional conflict (45) [5]. One previous study by Flynn et al., among patients who faced the decision whether to participate in phase I clinical trials also found clinically elevated scores (26 for trial acceptors and 34 for decliners) [9]. Thus, the DCS represents a clinically relevant tool that can be used to assess the quality of the decision whether to participate in early phase clinical cancer trials.

The factors that contribute to decisional conflict, specifically in the setting of contemplating early phase clinical trial participation, are largely unknown. Studies regarding standard cancer treatment decisions have indicated that, for instance, higher health-related quality of life (at baseline) [10] and lower health literacy [11] relate to more decisional conflict. However, whether this applies to decisions upon early phase clinical trial participation remains to be determined. In this particular context, hope might also be important to take into account, because it is often the main reason for patients to seek participation in early phase clinical trials [1,12,13]. A higher sense of hope may result in less decisional conflict. Moreover, patients who are satisfied with their (initial) consultation regarding early phase clinical trials might feel that their needs have been better met [14]. This could potentially result in less decisional conflict. Besides, a previous study found more decisional conflict in patients who decided not to participate in phase I clinical trials compared with patients who participated [9]. Lastly, previous reviews indicated that decisional conflict differed between the timing of measurement: studies that measured decisional conflict before and within one month after making a standard cancer treatment decision found higher levels compared with those who measured it later in time [5,8]. However, rather than the timing of measurement, the timing of decision making (i.e., the moment in time that patients made their decision) could matter as well. We expect that patients who make their decision sooner experience less decisional conflict. Untangling the contribution of the different variables that may affect decisional conflict will enable a better understanding of how to improve the quality of this difficult decision-making process.

In this study, we aim to provide a better understanding of decisional conflict in patients with advanced cancer who have recently been asked to decide whether to participate in early phase clinical cancer trials (i.e., phase I or phase I/II). We hypothesize that patients experience more decisional conflict if they have: (1) higher health-related quality of life, (2) lower health literacy, (3) lower sense of hope, (4) lower satisfaction with the initial consultation, (5) decided not to participate (rather than to participate), or (6) made the decision later in time. The insights from this study could support healthcare professionals or organizations in how to guide their patients through this difficult decision-making process.

## 2. Materials and Methods

### 2.1. Design

This study is part of a prospective project on decision making regarding participation in early phase clinical trials [15], registered in the Netherlands Trial Registry (NL7335). The Medical Ethics Committee of the Erasmus MC has reviewed the research proposal for the project (MEC-2018-151). Research governance approval was received from all participating hospitals, i.e., the Erasmus MC (Rotterdam, the Netherlands), the Netherlands Cancer Institute (Amsterdam, The Netherlands), and UMC Utrecht (Utrecht, The Netherlands). Yearly, these hospitals together see approximately 400–500 new patients who consider participation in early phase clinical trials.

### 2.2. Participants

Following the study protocol [15], the inclusion criteria were: diagnosed with advanced cancer and eligible for first participation in an early phase clinical trial (i.e., phase I or phase I/II), aged 18 years or older, sufficient command of the Dutch language (i.e., to be able to complete the questionnaires), and written informed consent. Exclusion criteria were: cognitive impairment according to the medical record, no access to the Internet to fill out the online questionnaires, or participation in another part of the project (*n* = 13).

### 2.3. Measurements

Participants completed two online questionnaires: a baseline questionnaire prior to their initial consultation with a medical oncologist regarding early phase clinical trial participation, and a final questionnaire three weeks after that consultation. Table 1 provides an overview of the measurements in the questionnaires. The primary outcome of decisional conflict was measured with the DCS [6,7,16], which has shown before to be a valid reliable outcome measure [8]. Other measurements were: health-related quality of life (subscales: global health status, physical functioning, role functioning, emotional functioning, cognitive functioning, social functioning, fatigue, nausea and vomiting, pain, dyspnea, insomnia, appetite loss, constipation, diarrhea, and financial difficulties) [17,18], health literacy [19,20,21], sense of hope [22,23], sociodemographic characteristics (age, gender, education level, nationality, living situation, and employment status), the satisfaction with the consultation, and the timing of the decision. Additional data regarding patient decision (i.e., decided to participate, decided not to participate) and medical situation (e.g., WHO status) were collected afterwards from electronic patient records by local trial monitors.

### 2.4. Procedure

Patients who were referred to the early phase clinical research units of the three hospitals between 18 February 2019 and 18 December 2020 were approached for participation. The study was put on hold between 16 March and 20 May 2020 due to restricting measures against the COVID-19 pandemic. Eligible patients were called by a nurse practitioner, trial secretary, or researcher from their own hospital to ask for first interest in this study. Patients who verbally agreed received an e-mail with the study information from a researcher (L.G.G.v.L.). The researcher then called the patient again to ask for preliminary consent. Patients who verbally consented received the baseline questionnaire via e-mail and were requested to complete it before their consultation with the medical oncologist. The questionnaire started by asking consent for participation in that particular questionnaire. Written informed consent was signed with the patient and collected before the start of the oncologist–patient consultation. Patients automatically received the final questionnaire three weeks later, which offered most patients sufficient opportunity to make a decision, but generally fell before the potential start of treatment in early phase clinical trials.

### 2.5. Statistical Analysis

Using IBM SPSS Statistics 25 descriptive statistics were generated for patient demographics, clinical characteristics, and individual measurements. Chi-square tests, Fisher’s exact tests, and a *t*-test were performed as non-response analyses to check for potential differences in gender, age, education level, living situation, and working situation between patients in the final analysis and patients who dropped out after giving written informed consent. Cronbach’s alphas were calculated to verify the internal reliability of health literacy, sense of hope, and decisional conflict. Univariate tests (i.e., independent t-test, one-way ANOVA with Tukey’s post hoc tests, or Pearson’s correlation analyses) were performed to explore relations for DCS scores with all other measurements. Eventually, a multiple linear regression analysis with backward selection was performed with decisional conflict as the dependent variable and using the measurements with a liberal level of significance (*p* ≤ 0.15) for their univariate analysis as independent variables [24].

## 3. Results

In total, 302 patients were asked to participate, of whom 227 (75.2%) gave preliminary consent and 193 (63.9%) gave written informed consent (see Figure 1). Of the 149 patients who completed the final questionnaire three weeks after the consultation (77.2% of all patients who gave written informed consent), 33 already knew that they did not meet the selection criteria for trial participation, for instance, because there were no available trials or because their health had deteriorated too much. These patients technically did not meet the inclusion criterion “‘eligible for first participation in an early phase clinical trial”. Moreover, the level of decisional conflict may have been affected by the fact that they could not make this decision themselves. We thus decided to exclude these patients from our analyses. Therefore, 116 patients were included in our analyses (60.1% of all patients who gave written informed consent) divided across 11 oncologists, who each saw 10–11 patients on average (*SD* = 11.5). Most patients were male (*n* = 78, 67.2%), of Dutch nationality (*n* = 115, 99.1%), lived with their partner (*n* = 73, 62.9%), and no longer had a job (*n* = 66, 56.9%). Table 2 provides an overview of patient characteristics; the few missing data are indicated in the table. Non-response analyses revealed no significant differences between the patients in the final analysis (*n* = 116) and the patients who dropped out after giving written informed consent (*n* = 77) in terms of gender (χ^2^(1) = 0.491, *p* = 0.483), age (*t*(191) = −0.503, *p* = 0.616), living situation (*p* = 0.323, Fisher’s exact test), and working situation (*p* = 0.290, Fisher’s exact test). There was a significant difference in education level (χ^2^(2) = 9.216, *p* = 0.010). Compared with the patients in the final analysis, patients who dropped out relatively more often had a low education level (41.6% vs. 25.0%) and less often a middle education level (20.8% vs. 39.7%), whereas high education levels were relatively similar (37.7% vs. 35.3%).

The baseline measurements in health literacy (α = 0.761) and sense of hope (α = 0.827), and the primary outcome decisional conflict (α = 0.902) all had an adequate to high reliability. In our sample (*n* = 116), the mean DCS score was 30.0 (*SD* = 16.9). Appendix A shows the boxplots for the DCS and subscales. All measurements and the significance level of their relations with decisional conflict are shown in Table 3 and Appendix A provides visualizations of significant relations. For categorical variables (i.e., decision and timing), the mean DCS score per group is indicated in Table 4. Appendix A shows all (cor)relations between measurements.

Significantly negative correlations were found for decisional conflict with two subscales of health-related quality of life, i.e., global health status (*r* = −0.322, *p* < 0.001) and social functioning (*r* = −0.200, *p* = 0.031), and significantly positive correlations with three subscales, i.e., fatigue (*r* = 0.208, *p* = 0.025), pain (*r* = 0.193, *p* = 0.038), and constipation (*r* = 0.214, *p* = 0.021). In other words, if patients had a better global health status and/or social functioning or experienced less fatigue, pain, and/or constipation, they reported less decisional conflict. Besides, health literacy (*r* = −0.202, *p* = 0.014) and sense of hope (*r* = −0.241, *p* = 0.003) showed significantly negative correlations with decisional conflict. Patients with a higher health literacy and/or a higher sense of hope experienced less decisional conflict. Furthermore, satisfaction with the consultation was significantly correlated with decisional conflict (*r* = −0.387, *p* < 0.001); patients who were more satisfied experienced less decisional conflict.

There was a significant relation between decisional conflict and patient decision (*t*(114) = −2.127, *p* = 0.036); patients who decided to participate (*M* = 27.7, *SD* = 17.1) experienced significantly less decisional conflict than those who decided not to participate (*M* = 34.6, *SD* = 15.7). Timing of the decision also had a significant relation with decisional conflict (*F*(115) = 16.135, *p* < 0.001); patients who made their decision before (*M* = 22.4, *SD* = 17.0) or within a week after the consultation (*M* = 26.3, *SD* = 14.1) experienced significantly less decisional conflict than patients who decided within 1–2 weeks (*M* = 43.1, *SD* = 7.5) or more than 2 weeks after the consultation (*M* = 45.8, *SD* = 11.0).

Based on the univariate analysis, the final multiple linear regression analysis was performed with eight subscales for health-related quality of life (i.e., global health status, physical functioning, social functioning, fatigue, pain, dyspnea, insomnia, and constipation), health literacy, sense of hope, satisfaction with the consultation, the decision, and timing of the decision. Table 5 shows that the ultimate regression model for decisional conflict with global health status, satisfaction with the consultation, and timing of the decision (reference category: more than 2 weeks after the consultation) as independent variables (also see Figure 2) proved significant (*F*(5,110) = 14.532, *p* < 0.001). A reasonable 37.0% of the variance in decisional conflict was explained by these variables with less decisional conflict reported by patients who had better global health status, were more satisfied with the initial consultation, and made the decision sooner.

## 4. Discussion

This study aimed to provide an understanding of decisional conflict in patients with advanced cancer who had recently been asked to decide whether to participate in early phase clinical trials (i.e., phase I or phase I/II). The hypothesized variables were indeed found to relate to decisional conflict in this population. In contrast to our hypothesis: (1) Patients who reported higher health-related quality of life, particularly in terms of a better global health status, reported less decisional conflict. In line with the other hypotheses, univariate analyses indicated that patients experienced more decisional conflict if they reported (2) lower health literacy, (3) lower sense of hope, (4) lower satisfaction with the initial consultation, (5) decided to not further consider participation (with clinically elevated levels similar to Flynn and colleagues [9]), and (6) made the decision more than one week after the consultation. The multivariable regression analysis showed that patients particularly experienced more (and clinically elevated) decisional conflict if they had a worse global health status, were less satisfied with the consultation, and made their decision more than one week after the consultation. Thus, our study opens various avenues towards improved decision making in the current context.

We were somewhat surprised that patients with lower global health status experienced more difficulties in deciding, as we expected lower global health status to be linked with the realization that trial participation is a less suitable option. However, for those patients the choice may represent weighing a further decrease of their health-related quality of life against a slim chance of benefit, where even the slightest chance of improving the outcome may seem appealing. In line with the potential patient intrinsic characteristics determining decisional conflict, we found that patients who made the decision sooner experienced less decisional conflict. The moment that patients make a decision could result from their decisional preferences. For instance, a patient’s decision to participate may relate to the extent that they want to reduce cognitive dissonance. Cognitive dissonance reduction is a psychological adaptation process in which threatening information, e.g., fatal prognosis, is (unintentionally) discarded [25]. Patients who make the decision sooner may possess this, which could improve their decisiveness and thereby reduce decisional conflict. Another explanation for this relation could be that patients who were more convinced about their choice (and thus experienced less decisional conflict) needed less time to make a decision. If patients need more time to make a decision, this could be reason for healthcare professionals—not only oncologists, but also those who specialize in discussing psycho-oncological matters—to provide additional support in resolving remaining issues. Future studies may elucidate how patients can be optimally supported in addressing such issues. Based on our study, both patient global health status at first consultation and decisiveness indicate whether more support is required.

Whereas patient intrinsic characteristics may be difficult to alter, a factor that oncologists can truly impact is the satisfaction with the consultation. Patients who were less satisfied with their consultation experienced more decisional conflict, which suits the idea that they felt their needs had not been sufficiently met during the consultation [14]. This seems to suggest that effective patient–provider communication can prevent or lower decisional conflict, although the reverse (that patients who report less decisional conflict also feel better about the consultation regardless of its content or helpfulness) also seems plausible. Higher satisfaction with the consultation could potentially be reached through patient-centered communication [26,27]. However, the complexity of early phase clinical trials can lead to a focus on medical–technical information with only limited discussion of patient values and preferences [28,29]. Indeed, we found that patients experienced most unresolved needs regarding the “values clarity” and “uncertainty” subscales. Extending the discussion on patient values could not only be beneficial to patient satisfaction and decisional conflict, but also to address specific needs of patients with, e.g., a lower global health status. Further research is needed to confirm whether exploring patient values decreases decisional conflict. The univariate analysis also points towards an important realization to have more attention for those patients who consider or decide not to participate. Although this group is usually referred for symptom-oriented care, previous studies showed that palliative care and patient prognosis are not always discussed during the deliberation of early phase clinical trial participation [30,31]. Patients may benefit from more information during the decision-making process about this option that they seriously consider. Although patient intrinsic characteristics are thus important, oncologists themselves also have an important role in preventing decisional conflict, particularly by (further) optimizing their consultations.

It is important to be aware of several biases that may underlie this study’s measurements. Since not all eligible patients consented to participate and many dropped out, the population that oncologists see regarding early phase clinical trials may be more diverse than our sample. First, health-related quality of life was better in this study compared to a general group of patients with stage III/IV cancer or with recurrent/metastatic cancer [32]. This difference is understandable, since patients need to be in good condition to even have the choice to participate in early phase clinical trials as illustrated by the exclusions from this study due to deteriorated health or death. As our patients form a relatively frail group, the dropout rate in this study is expected. Second, the non-response analysis showed that relatively many patients with a low education level dropped out from this study. Moreover, we included only one patient below the cut-off for low health literacy and one with a non-Dutch nationality in the final analysis. Although sufficient command of the Dutch language was an inclusion criterion, these numbers differ strikingly from the 36% people with low health literacy [33] and 25% with a migration background [34] in the Netherlands. This could explain why health literacy was excluded from the regression model. Possibly, early phase clinical trial participation was mainly suggested to people with high health literacy (with generally higher levels of education [35]) and of Dutch nationality, because their treating oncologists may have (correctly [36]) believed they would be better able to understand complex trial information. The univariate analysis indeed indicated that patients with an even better health literacy were better able to make a decision. Presenting trial information in a simple comprehensive manner could thus be helpful to everybody with lower health literacy and empower them to consider trial participation; moreover, simplification could be beneficial to all [36]. Third, similar to a previous study [25], patient sense of hope was quite high in this study. Hope appears important to virtually all patients with advanced cancer [37], therefore it seems understandable why this factor was removed from the regression model. Last, because our study was based on fairly limited available previous data regarding decisional conflict, especially in the context of deciding on early phase clinical trial participation, there may be other factors that codetermine decisional conflict. For instance, we did not investigate individual differences between oncologists. Our study provides an initial input for developing new theories and models into decisional conflict or decision making in general.

## 5. Conclusions

In conclusion, the main determinants of decisional conflict are either patient intrinsic, i.e., global health status and timing of the decision, or affected by interaction with oncologists, i.e., satisfaction with the consultation. Although intrinsic characteristics may be difficult to alter when aiming to prevent decisional conflict, oncologists still have an important role to play. Future research should verify the suggestion that exploring patient values and preferences can positively affect patient satisfaction with the consultation, thereby improving the quality of the decision-making process and eventually patient quality of life.

## Figures and Tables

**Figure 1 cancers-14-01500-f001:**
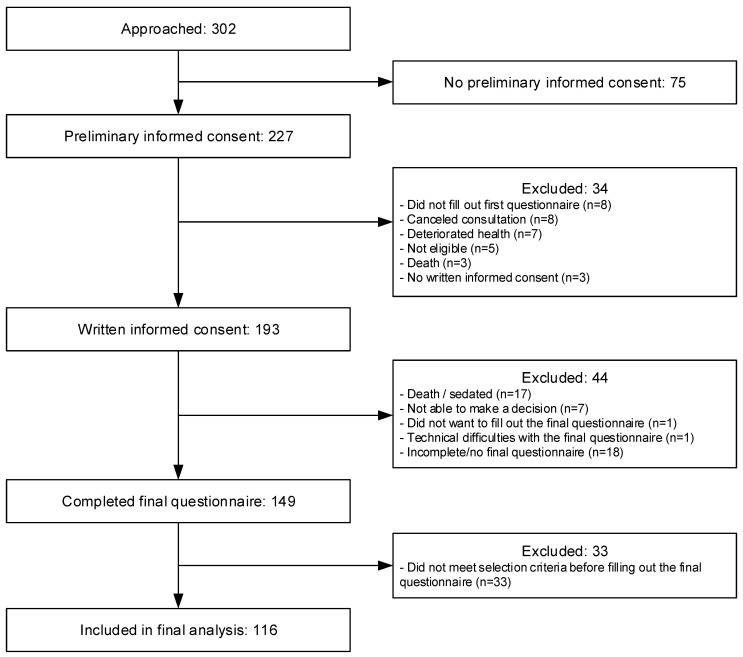
Flowchart for the inclusion of patients.

**Figure 2 cancers-14-01500-f002:**
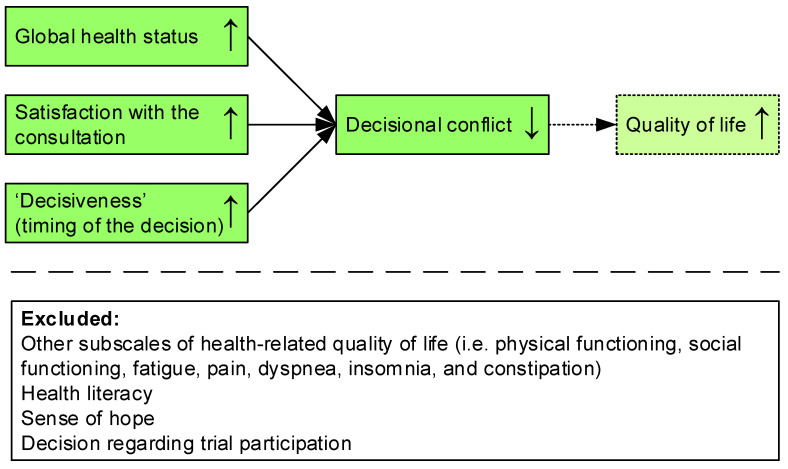
Model of predictors for decisional conflict.

**Table 1 cancers-14-01500-t001:** Overview of measurements.

Outcome	Instrument
Primary outcome (final questionnaire)	
Decisional conflict	The validated Dutch version [16] of the Decisional Conflict Scale (DCS) [6,7] was used. The DCS consists of 16 items on a 5-point Likert scale (0 = strongly agree, 4 = strongly disagree) within the informed (3 items), values clarity (3 items), support (3 items), uncertainty (3 items), and effective decision (4 items) subscales. In line with the user manual [7], all items were summed, divided by 16, and multiplied by 25 to get a total score for decisional conflict on a 0–100 scale. Scores below 25 are associated with being able to implement a decision and those above 37.5 with decision delay.
Baseline measurements (baseline questionnaire)	
Quality of life	Quality of life was assessed with the QLQ-C30 version 3.0 [17] in Dutch of the European Organization for Research and Treatment of Cancer (EORTC). The QLQ-C30 consists of subscales/items for global health status (2 items), physical functioning (5 items), role functioning (2 items), emotional functioning (4 items), cognitive functioning (2 items), social functioning (2 items), fatigue (3 items), nausea and vomiting (2 items), pain (2 items), dyspnea (1 item), insomnia (1 item), appetite loss (1 item), constipation (1 item), diarrhea (1 item), and financial difficulties (1 item). Global health status was assessed on a 7-point Likert scale (1–7), whereas all functional and symptom scales/items were assessed on a 4-point Likert scale (1–4). In line with the EORTC manual [18], linear transformation was applied to obtain a score between 0–100 for all scales and items.
Health literacy	Patient ability to perform the basic reading and numerical tasks required to function in the health care environment was assessed using the 3-item Set of Brief Screening Questions (SBSQ) [19] on a 5-point Likert scale (1–5). We used the previously translated Dutch version (SBSQ-D) [20,21]. Based on previous literature, health literacy was considered low if the mean of the three items was ≤2, and adequate for >2.
Sense of hope	The Herth Hope Index (HHI) [22] was used to measure a global non-time-oriented sense of hope. The 12 items have a 4-point Likert scale (1–4) and were previously translated and authorized in Dutch [23]. Since it was advised for the Dutch version to use the scale as a whole rather than using subscales, the items were summed to a total score within a range between 12–48.
Other measurements (final questionnaire)	
Satisfaction with the communication	Satisfaction with the consultation was measured with one question (“How satisfied were you with the initial consultation?”) that could be assessed on a 7-point Likert-scale (1 = completely unsatisfied, 7 = completely satisfied).
Timing of the decision	Timing of the decision was measured with one question (“When did you approximately decide to participate or not in an early phase clinical trial?”) with 5 answer possibilities (1 = before initial consultation, 2 = within 1 week after the initial consultation, 3 = within 1–2 weeks after the initial consultation, 4 = more than 2 weeks after the initial consultation, 5 = not yet). For the analysis, the latter two possibilities were combined.

**Table 2 cancers-14-01500-t002:** Patient characteristics (*n* = 116).

Patient Characteristics	*M* (*SD*) or *n* (%)
Gender, *n* (%)	
Female	38 (32.8)
Male	78 (67.2)
Age, *M* (*SD*)	62.5 (8.8)
Education level, *n* (%)	
Low (no education to lowest level of secondary education)	29 (25.0)
Middle (senior general secondary and pre-university education)	46 (39.7)
High (higher vocational education and university)	41 (35.3)
Nationality, *n* (%)	
Dutch	115 (99.1)
Other	1 (0.9)
Living situation, *n* (%)	
Alone	12 (10.3)
With partner	73 (62.9)
With partner and child(ren)	27 (23.3)
With child(ren) or other relative(s)	4 (3.4)
Working situation, *n* (%)	
Paid job	32 (27.6)
No job (anymore)	66 (56.9)
In health insurance act	10 (8.6)
Other (e.g., voluntary work)	8 (6.9)
Hospital, *n* (%)	
Erasmus MC	84 (72.4)
Netherlands Cancer Institute	23 (19.8)
UMC Utrecht	9 (7.8)
WHO performance status at initial consultation, *n* (%)	
0	27 (23.3)
1 or 2	80 (69.0)
Missing/unknown	9 (7.8)
Primary diagnosis, *n* (%)	
Colorectal/anal cancer	35 (30.2)
Esophageal/stomach cancer	11 (9.5)
Hepatobiliary/pancreatic cancer	20 (17.2)
Gynecological cancer	6 (5.2)
Lung cancer/mesothelioma	8 (6.9)
Urinary tract cancer	26 (22.4)
Breast cancer	3 (2.6)
Melanoma/skin cancer	3 (2.6)
Other	4 (3.4)
Metastases, *n* (%)	
Yes	110 (94.8)
No	6 (5.2)
Number of previous lines of therapy, *M* (*SD*)	2.6 (1.6)
Missing/unknown, *n* (%)	6 (5.2)
Participated in another (phase II/III) clinical trial, *n* (%)	
Yes	26 (22.4)
No	90 (77.6)

**Table 3 cancers-14-01500-t003:** Patient measurements (*n* = 116). *p*-values are provided for the univariate relation with decisional conflict.

Patient Measurements	*M* (*SD*) or *n* (%)	*p*-Value
Decisional conflict (DCS), *M* (*SD*)		
Informed subscale	31.5 (21.3)	N/A
Values clarity subscale	36.6 (20.6)	N/A
Support subscale	24.2 (18.1)	N/A
Uncertainty subscale	36.6 (24.9)	N/A
Effective decision making subscale	23.3 (20.4)	N/A
Total decisional conflict	30.0 (16.9)	N/A
Quality of life (QLQ-C30), *M* (*SD*)		
Global health status		
Global health status	67.4 (18.4)	<0.001 ^‡^
Functional scales		
Physical functioning	80.1 (16.8)	0.052 ^‡^
Role functioning	68.4 (24.6)	0.105 ^‡^
Emotional functioning	76.3 (18.6)	0.190 ^‡^
Cognitive functioning	87.2 (18.8)	0.377 ^‡^
Social functioning	78.7 (23.8)	0.031 ^‡^
Symptom scales/items		
Fatigue	28.6 (20.8)	0.025 ^‡^
Nausea and vomiting	8.2 (15.5)	0.822 ^‡^
Pain	22.4 (24.5)	0.038 ^‡^
Dyspnea	13.5 (21.1)	0.077 ^‡^
Insomnia	19.3 (26.8)	0.099^‡^
Appetite loss	17.8 (25.4)	0.655 ^‡^
Constipation	12.9 (24.0)	0.021 ^‡^
Diarrhea	11.5 (19.2)	0.869 ^‡^
Financial difficulties	7.2 (16.9)	0.334 ^‡^
Health literacy, *M* (*SD*)	4.5 (0.6)	0.023 ^‡^
Sense of Hope (HHI), *M* (*SD*)	36.8 (4.9)	0.003 ^‡^
Satisfaction with the consultation, *M* (*SD*)	6.0 (1.3)	<0.001 ^‡^
Missing, *n* (%)	1 (0.7)	
Decision regarding trial participation, *n* (%)		0.036 ^┴^
Decided to further consider participation	77 (66.4)	
Decided to not further consider participation (and not to participate)	39 (33.6)	
Timing of the decision, *n* (%)		<0.001 ^†^
Before initial consultation	38 (32.8)	
Within 1 week after the initial consultation	47 (40.5)	
1–2 weeks after the initial consultation	9 (7.8)	
More than 2 weeks after the initial consultation	22 (19.0)	

^‡^—*p*-value from correlation analysis; ^┴^—*p*-value from t- test; ^†^—*p*-value from ANOVA.

**Table 4 cancers-14-01500-t004:** Mean DCS score per group for the categorical variables (i.e., decision and timing of the decision).

	DCS Score *M* (*SD*)
Decision regarding trial participation	
Decided to further consider participation	27.7 (17.1)
Decided to not further consider participation (and not to participate)	34.6 (15.7)
Timing of the decision	
Before initial consultation	22.4 (17.0)
Within 1 week after the initial consultation	26.3 (14.1)
1–2 weeks after the initial consultation	43.1 (7.5)
More than 2 weeks after the initial consultation	45.8 (11.0)

**Table 5 cancers-14-01500-t005:** Linear model of predictors of DCS scores.

	*F*	df	Adjusted R2	*p*-Value
Model statistics	14.532	(5, 110)	0.370	<0.001
	** *b* **	**95% CI**	**β**	***p*-value**
Global health status	−0.170	(−0.311, −0.030)	−0.185	0.018
Satisfaction with the consultation	−3.153	(−5.117, −1.189)	−0.246	0.002
Timing of the decision *				
Before initial consultation	−19.445	(−26.812, −12.079)	−0.543	<0.001
Within 1 week after the initial consultation	−14.644	(−21.900, −7.388)	−0.427	<0.001
1–2 weeks after the initial consultation	−2.047	(−12.588, 8.494)	−0.033	0.701
Constant	72.798	(58.826, 86.769)		<0.001

* The reference category for timing of the decision was “more than 2 weeks after the consultation”.

## Data Availability

The data presented in this study are available on reasonable request from the corresponding author. The data are not publicly available due to privacy and ethical considerations.

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
