# Peer review of "Decisional Conflict after Deciding on Potential Participation in Early Phase Clinical Cancer Trials: Dependent on Global Health Status, Satisfaction with Communication, and Timing"

_cancers, 2022, doi:10.3390/cancers14061500_

Round 1

Reviewer 1 Report

Thank you for the opportunity to review this manuscript about decisional conflict in participation in early clinical trial.

I think the topic is interesting and important for cancer care – but I think the manuscript could be improved in communicating. The results would be easier to grasp in figures. Especially the abstract must be improved. I am also bit concerned about the phrasing “early phase”. I would recommend to change to " Phase I/II studies" in the title and throughout the manuscript since this is a more conventional phrasing that all cancer researcher and oncologist know what it is.  It also has to be clearer from the beginning that this study is performed in “patients with advanced cancers”.

Abstract: The reader do not get a good overview over the study in the Abstract. Please add clearly which kind of patients that participated i.e. patient with advanced cancer,  that the design of the study is a prospective study with questionnarire etc. Add more clear results in the study with values and p-valus or 95%CI. To present the results with values for at least the primary endpoint is a minimum. Although I know it is challenging to have only 200 word to use for this I think the authors should remove a lot of the background and just keep the aim of the study, short down the conclusion to one sentence and future direction is not necessary in the abstract. The focus should be on the current study, its design and the results.

Introduction: I lack the context of patients in palliative phase and the challenges in performing clinical trials in palliative care patients.

Table1 : Very difficult to read and grasp the content. Please improve the layout. E.g. right-adjusted columns and bullet points for the outcomes.

Table 2: Difficult to read. Right-adjust the first column.

Results: The results about the correlation between the different variables would be much easier to understand and grasp for the reader if you present it as figures. E.g. scatter plots with regressions-lines. I recommend that at least the primary endpoint should be presented in that way but preferable also other results that the authors want to high-light. Results presented only as r-values and p-values is more difficult to grasp.

Discussion: Well-written. The first sentence in the discussion could be used in the abstract as the first sentence but changed to Phase I/II studies.

Conclusion: Well-written and informative.

Author Response

I think the topic is interesting and important for cancer care – but I think the manuscript could be improved in communicating. The results would be easier to grasp in figures. Especially the abstract must be improved. I am also bit concerned about the phrasing “early phase”. I would recommend to change to " Phase I/II studies" in the title and throughout the manuscript since this is a more conventional phrasing that all cancer researcher and oncologist know what it is.

We wish to thank reviewer 1 for his/her kind words regarding our manuscript and suggestions. With regard to the results and abstract, please see the corresponding comments below. With regard to the phrasing, we now operationalize “early phase clinical trials” as “phase I or phase I/II clinical trials” in line 105 of the introduction and in line 248 of the discussion, as we already did in our methods section (line 123-124). We believe that this makes this operationalization more clear throughout the manuscript. However, we wish to deviate from the suggestion to change “early phase clinical trials” into “phase I/II clinical trials” in the title (and other places) based on several reasons. First, a quick search in PubMed revealed 22,866 results for “Early phase clinical trials” compared to 6,556 for “phase I/II clinical trials”, which suggests that the first (instead of the latter) is the more conventional phrasing. Second, our study did not only include patients who were referred for phase I/II clinical trials, but also for phase I clinical trials. Thus, a change to “phase I/II clinical trials” (nor to “phase I clinical trials”) would not suffice in this case, whereas “early phase clinical trials” does suffice.

It also has to be clearer from the beginning that this study is performed in “patients with advanced cancers”.

We have now clarified that the study focuses on “patients with advanced cancer” in line 100 of the introduction (i.e. where we describe the study aim) to make this more clear early on in the manuscript.

Abstract: The reader do not get a good overview over the study in the Abstract. Please add clearly which kind of patients that participated i.e. patient with advanced cancer,  that the design of the study is a prospective study with questionnarire etc. Add more clear results in the study with values and p-valus or 95%CI. To present the results with values for at least the primary endpoint is a minimum. Although I know it is challenging to have only 200 word to use for this I think the authors should remove a lot of the background and just keep the aim of the study, short down the conclusion to one sentence and future direction is not necessary in the abstract. The focus should be on the current study, its design and the results.

We have now added specifications regarding the kind of patients that participated (line 22, 31-32 and 35) and that the design is a “prospective multicenter study with questionnaires” (line 31). The mean and standard deviation for decisional conflict are indicated in line 41. We have added β- and p-values to help illustrate the relation of the three variables in the multivariate regression analysis with decisional conflict (line 38-39). Following reviewer 1’s suggestion, we have shortened the background to free up necessary words for the above-mentioned changes (line 28-35). We did not shorten the conclusion, as we believe that this helps readers who are less familiar with multivariate regression analysis to understand the meaning and potential implications of our results. Vice versa, readers who are familiar with multivariate regression analysis will most likely prefer the complete set of values for this analysis as reported in Table 5 in the results section.

Introduction: I lack the context of patients in palliative phase and the challenges in performing clinical trials in palliative care patients.

Regretfully, we are uncertain what the reviewer means with this comment. In our introduction, we point out that ‘a well-balanced decision-making process is essential to ensure that these patients spend the limited time towards the end of their lives in line with their personal values and beliefs’ (line 59-61). With this remark, we indicate that patients life expectancy is short and we emphasize the importance of integrating these patients’ personal values, which aligns well with the importance given to patient autonomy in palliative care. Furthermore, we are uncertain if reviewer 1 refers to challenges in performing clinical trials in general, or with regards to our current study. In case of the first, we believe this lies beyond the scope of our manuscript, since we focus on the preceding decision-making process rather than the outcome of that process (and thus the challenges of trial participation). In case of the latter, conducting clinical trials in palliative care patients is indeed a challenge and we are very proud to have included so many patients facing this decision in our study, which could be explained by the fact that our study only entailed little extra burden (i.e. two short questionnaires) for these patients.

Table1 : Very difficult to read and grasp the content. Please improve the layout. E.g. right-adjusted columns and bullet points for the outcomes.

We have left-aligned (instead of centred) the information in Table 1 to improve readability and to make the distinction between the variables clearer/better visible.

Table 2: Difficult to read. Right-adjust the first column.

We have left-aligned (instead of centred) the left column and increased indent for the categories of categorical variables in Table 2 to improve readability. We have also applied the same strategy to tables 3, 4 and 5.

Results: The results about the correlation between the different variables would be much easier to understand and grasp for the reader if you present it as figures. E.g. scatter plots with regressions-lines. I recommend that at least the primary endpoint should be presented in that way but preferable also other results that the authors want to high-light. Results presented only as r-values and p-values is more difficult to grasp.

The figures regarding the univariate (cor)relations for the variables with the main outcome decisional conflict were already included as supplementary, i.e. Figure S2, which we refer to in line 203 and is included in the manuscript-supplementary.zip. Because this figure shows the relation between decisional conflict and six other variables, it has become quite large – possibly too large to be ‘printed’ in sufficient quality. If the editor agrees with reviewer 1, we can add it to the main manuscript/as a (non-supplementary) figure. For now, we have left the figure as a supplementary.

Discussion: Well-written. The first sentence in the discussion could be used in the abstract as the first sentence but changed to Phase I/II studies.

Thank you for this compliment and suggestion. We now also use a similar sentence as the third sentence of our abstract (this sentence is only preceded by two short sentences to explain the setting and the relevance of our study respectively).

Conclusion: Well-written and informative.

Thank you for this compliment.

Reviewer 2 Report

The decisions that patients are making through the course of the cancer disease up to the palliative care are of the highest importance and are influencing without any doubt the patients' and their families quality of life. The decision whether to participate or decline the clinical trial is fitting in the picture. In my opinion every study which can bring something new to the topic of what influencing the patient's decision making pattern is of the highest importance too. 

The study presented in the manuscript is well prepared and conducted, beeing a part of a bigger project. 

In my opinion the statistical tools are selected correctly.

The results are well described and supported by the tables. The conclusions are well supported by the results.

In the Chapter of Discussions Authors are highlighting the possibble biases and weaknesses of the study, indicating that it is just a preliminary study guiding the direction for further investigations but on the other hand, also supportive to the clinicians during consultations with the patients.

Author Response

The decisions that patients are making through the course of the cancer disease up to the palliative care are of the highest importance and are influencing without any doubt the patients' and their families quality of life. The decision whether to participate or decline the clinical trial is fitting in the picture. In my opinion every study which can bring something new to the topic of what influencing the patient's decision making pattern is of the highest importance too. 

The study presented in the manuscript is well prepared and conducted, beeing a part of a bigger project. 

In my opinion the statistical tools are selected correctly.

The results are well described and supported by the tables. The conclusions are well supported by the results.

In the Chapter of Discussions Authors are highlighting the possibble biases and weaknesses of the study, indicating that it is just a preliminary study guiding the direction for further investigations but on the other hand, also supportive to the clinicians during consultations with the patients.

We wish to thank reviewer 2 for his/her positive feedback.

Reviewer 3 Report

MS Title: Decisional Conflict after Deciding on Potential Participation in Early Phase Clinical Cancer Trials: Dependent on Global Health Status, Satisfaction and Timing

This Manuscript is interesting because the issues of the Decisional Conflict after Deciding on Potential Participation in Early Phase Clinical Cancer Trials are addressed in it a very insightful way, which can be useful to both clinicians and researchers involved in oncology care. Moreover, it is possible that in the future, findings from this study will contribute to the improvement of the decision-making process for many patients with end-stage cancer, leading to a better quality of life in this vulnerable group.

Suggestions for some minor changes are listed below.

P # 2 Introduction

It is:

The DCS has been used regularly for cancer patients[8], but predominantly for standard treatment decisions.

 It should be:

The DCS has been used regularly for patients with cancer [8], but predominantly for standard treatment decisions.

P # 12 Conclusions

In the Conclusions, the Authors may briefly comment on a possible role of consultations by psycho-oncologists, as potential facilitating factors for the patients’ decision-making process. Perhaps, this option could also be investigated in future research studies.

Author Response

This Manuscript is interesting because the issues of the Decisional Conflict after Deciding on Potential Participation in Early Phase Clinical Cancer Trials are addressed in it a very insightful way, which can be useful to both clinicians and researchers involved in oncology care. Moreover, it is possible that in the future, findings from this study will contribute to the improvement of the decision-making process for many patients with end-stage cancer, leading to a better quality of life in this vulnerable group.

Suggestions for some minor changes are listed below.

P # 2 Introduction

It is: The DCS has been used regularly for cancer patients[8], but predominantly for standard treatment decisions. It should be: The DCS has been used regularly for patients with cancer [8], but predominantly for standard treatment decisions.

P # 12 Conclusions

In the Conclusions, the Authors may briefly comment on a possible role of consultations by psycho-oncologists, as potential facilitating factors for the patients’ decision-making process. Perhaps, this option could also be investigated in future research studies.

We are grateful for reviewer 3’s compliments. In accordance with the remarks of reviewer 3, we have changed “cancer patients” to “patients with cancer” (line 74). Furthermore, we have added a brief comment on the possible facilitating role of healthcare professionals in the psycho-oncological field (line 277-282).

Round 2

Reviewer 1 Report

I think the manuscript is improved after the revision. All my questions have been adressed in a good way.  I am sorry that I missed the supplementary files with the informative figures in my previous review. I am happy that they are included as supplementary files or in the manuscript as the Editor decide.